# Introducing platform surface interior angle (PSIA) and its role in flake formation, size and shape

Shannon P. McPherron[1]*, Aylar Abdolahzadeh[2], Will Archer[1,3], Annie Chan[8,10], Igor Djakovic[4], Tamara Dogandžić[1], George M. Leader[2,7], Li Li[5], Sam Lin[6], Matthew Magnani[9], Jonathan Reeves[5], Zeljko Rezek[1], Marcel Weiss[1]

**1** Department of Human Evolution, Max Planck Institute for Evolutionary Anthropology, Leipzig, Germany, **2** Department of Anthropology, University of Pennsylvania, Philadelphia, Pennsylvania, United States of America, **3** Department of Archaeology and Anthropology, National Museum, Bloemfontein, South Africa, **4** Department of Archaeology, University of Leiden, Leiden, The Netherlands, **5** Department of Early Prehistory and Quaternary Ecology, Eberhard Karls University of Tübingen, Tübingen, Germany, **6** Centre for Archaeological Science, School of Earth, Atmospheric and Life Sciences, University of Wollongong, Wollongong, Australia, **7** Department of Sociology and Anthropology, The College of New Jersey, Ewing, New Jersey, United States of America, **8** Department of Asian Studies, Ludwig-Maximilians-Universität München, Munich, Germany, **9** Department of Anthropology, Harvard University, Cambridge, Massachusetts, United States of America, **10** Centre de Recherche sur les Civilisations de l'Asie Orientale, Paris, France

* mcpherron@eva.mpg.de

**Data Availability Statement:** All relevant data are within the manuscript and its Supporting information files.

## Abstract

Four ways archaeologists have tried to gain insights into how flintknapping creates lithic variability are fracture mechanics, controlled experimentation, replication and attribute studies of lithic assemblages. Fracture mechanics has the advantage of drawing more directly on first principles derived from physics and material sciences, but its relevance to controlled experimentation, replication and lithic studies more generally has been limited. Controlled experiments have the advantage of being able to isolate and quantify the contribution of individual variables to knapping outcomes, and the results of these experiments have provided models of flake formation that when applied to the archaeological record of flintknapping have provided insights into past behavior. Here we develop a linkage between fracture mechanics and the results of previous controlled experiments to increase their combined explanatory and predictive power. We do this by documenting the influence of Herztian cone formation, a constant in fracture mechanics, on flake platforms. We find that the platform width is a function of the Hertzian cone constant angle and the geometry of the platform edge. This finding strengthens the foundation of one of the more influential models emerging from the controlled experiments. With additional work, this should make it possible to merge more of the experimental results into a more comprehensive model of flake formation.

## Introduction

There is considerable literature dedicated to understanding how flakes form. This understanding has come from work following several broad approaches including fracture mechanics

**Funding:** We thank the Max Planck Society for funding portions of this work. LL and JR received funding from the European Research Council (ERC) under the European Union's Horizon 2020 research and innovation program (grant agreement n° 714658; STONECULT project). TD received funding from the European Union's Framework Programme for Research and Innovation Horizon 2020 (2014-2020) under the Marie Skłodowska-Curie grant agreement No. 751125. Specifically: SPM, MW, WA, and ZR receive salary from the Max Planck Society through the Max Plank Institute for Evolutionary Anthropology, the Department of Human Evolution. The department also paid equipment, travel money, and other expenses that allowed this study to be done. LL and LR have their salaries paid by European Research Council (ERC) under the European Union's Horizon 2020 research and innovation program (grant agreement n° 714658; STONECULT project. TD has her salary paid by her grant from the European Union's Framework Programme for Research and Innovation Horizon 2020 (2014-2020) under the Marie Skłodowska-Curie grant agreement No. 751125. The funders had no role in study design, data collection and analysis, decision to publish, or preparation of the manuscript.

**Competing interests:** The authors have declared that no competing interests exist.

(e.g. [1–4]), controlled experiments (e.g. [5–13]), replicative experiments (e.g. [14–16]), and attribute analysis of archaeological assemblages. These approaches each have their own strengths and weaknesses; however, one way to think about the differences among them is in the directionality of inference. Fracture mechanics starts with first principles, or laws drawn from physics and material sciences in particular, concerning how fractures should form in brittle solids. These are then used to make predictions, typically with explicit mathematical models, about how flakes should look (size and shape) under varying conditions of force application and solid material properties. These predictions are then typically tested experimentally. To the contrary, controlled and replicative experiments and studies of actual lithic assemblages look at empirical regularities in flake size and shape under varying conditions of core preparation and flaking (where the core is struck, the hammer type, how the platform is prepared, the angle of strike, etc.). The observed outcomes are then summarized as statistical models of flake formation. From these models one can try to infer first principles, but for reasons discussed next, this is usually not done. All of these approaches to understanding flake formation are, of course, valid and useful. Ideally, what is learned from actually doing (experiments) and what is learned from knowing how it should work in principle (fracture mechanics) inform each other in an iterative loop.

Understanding first principles causality from statistical modeling, however, is challenging. McElreath [17] gives the example of trying to understand the physics behind race cars by measuring their attributes. Knowing the speed and handling characteristics of each car, eventually the right things could be measured on each car to build statistical models with enough predictive power to know how a new car might perform in a race, but it would be quite difficult to infer the general physical concepts (and laws) like torque, angular momentum, friction, aerodynamics, conservation of energy, etc. from these statistical models. Of course, with prior knowledge of the physics, finding the right attributes to measure on the cars and statistical modeling are more quickly and accurately done. This is important because even when the physical laws are known, modeling them directly can be prohibitively complex or computationally expensive (e.g. air resistance) whereas experiments and statistical models can more efficiently arrive at useable solutions.

The same is true of studies of flake formation and the role of fracture mechanics within it. Fracture mechanics itself is a longstanding and widely-applied field of study, but its practical application has been extremely limited in our field, with the best examples coming from the papers of Cotterell and Kamminga [2, 3] and of Speth [4, 5]. These papers start with the physics of fracture mechanics in brittle solids to then explain how flakes are formed and, therefore, why they vary. Some attributes, like the bulb of percussion, are more easily accounted for directly in fracture mechanics (e.g. Hertzian cone formation), whereas for other attributes, like flake size and shape, the conceptual and mathematical frameworks are provided by fracture mechanics. However, as with the just mentioned example of air resistance, translating the physics of how flakes are formed into a workable model that can predict flake size and shape given the relevant parameters (e.g. core shape, angle of blow, force of blow, etc.) has not come to pass (cf. [4]), and it may not come to pass any time soon.

So instead, while some papers on controlled experiments in flake formation may cite studies from fracture mechanics, their approaches are all based on statistical modeling of the empirical relationships between variables of flaking and flake outcomes. Speth's work on this topic is a good example. His 1972 paper uses fracture mechanics to derive a formula to predict flake size which is then tested against a set of actual flakes from a prehistoric site. By 1975 and again in 1981, Speth had moved to experimental approaches (ball bearings on glass) and the connection back to fracture mechanics had all but disappeared. Dibble [10] goes further and dismisses fracture mechanics from the start as nearly irrelevant. Instead of looking to fracture mechanics

for insights into what to study, experimental studies are being informed by replicative knappers and observations on how actual lithic assemblages vary. Dibble [10, 18] is explicit in stating that his experimental research is based on what knappers would have been able to control. In these experiments, coming back to the race car analogy, we are carefully building cars controlling for engine size, wheel configurations, foils, etc., things that are generally thought to be important for making a car go fast, and then measuring their speeds. Again, though, because it is also difficult to go in the other direction (from statistical modeling back to first principles), the controlled experiment papers have not produced a general model of how flakes form. Instead, we have a series of statistical models that are difficult to relate to one another (e.g. [19]). Part of the difficulty stems from the fact that different knapping variables are likely related in complex ways during flake formation, yet their effects on flake variation are limited to a similar set of size and shape attributes. For instance, Magnani et al. [19] showed that changing the angle of blow, the location of percussion and hammer hardness all result in similar flake attribute variation. While regression models can capture the influence of these individual knapping factors, reconciling them into a causal sequence of flake formation remains challenging without the guidance of a general model.

The strongest and most influential model derived from controlled experiments is the exterior platform angle and platform depth (EPA-PD) model. The EPA-PD model states that flake size (weight) is primarily a function of two important variables: exterior platform angle (EPA) and platform depth (PD) (Fig 1). Increasing either of these variables will increase flake size, but the relationship between the two is geometric such that at higher values of EPA changes in PD have a greater effect on flake size. The EPA-PD model has been replicated in multiple ways, including experiments in the material sciences [6–8, 10, 20, 21] and in actual lithic assemblages (e.g. [18, 22, 23]). It is also argued that in certain conditions, EPA-PD has a stronger effect on flake size and shape than does core surface morphology [11, 24]. The EPA-PD model of flake formation, however, is constrained in what it can explain. For instance, beveled flakes, where the volume behind the platform is thinned, are not easily included into the model [25]. Beveled flakes are typically larger (weight) than the EPA-PD models predicts given their lower platform depths. The EPA-PD model also does not explain why flake size and shape change with varying angles of blow and platform shape [9, 19, 26]. It also does not account for flake width, which is obviously a major component of shape. It is worth noting that while the percentage of variability in weight explained by the EPA-PD model ($R^2$) is typically high in the Dibble glass experiments, it is far lower in actual lithic assemblages [26]. It is low enough that its utility for measuring retouch intensity (i.e. knowing how much mass has been removed from a flake through retouch) is limited [14, 27–29]. As a result, there have been various proposals to improve the statistical modeling of flake weight (or size) from different sets of measures (e.g. [28, 30–32]). Again, though, without a general model, it is not really clear why one measurement technique should work better than another. Because of this, the success of these models is measured by the amount of empirical variation the models can account for (i.e. $R^2$ values) rather than against theoretical predictions. This approach reflects the fundamental issue that our knowledge of how flake formation works is still too limited to be translated into measurable attributes. Instead, the majority of the lithic attributes commonly measured by archaeologists are derived from intuitive observations through actualistic flint-knapping (i.e., when we do x, flakes tend to show y). As Speth [4] noted nearly a half century ago, how or even whether these attributes are indeed meaningful for explaining flake variation are typically not justified *a priori* by theoretical models of flake formation.

Here we propose to build on the EPA-PD model by 1) switching the focus from variables controlled by the knapper to variables that might be more directly related to flake initiation and formation and by 2) drawing insights from the fracture mechanics literature. In particular,

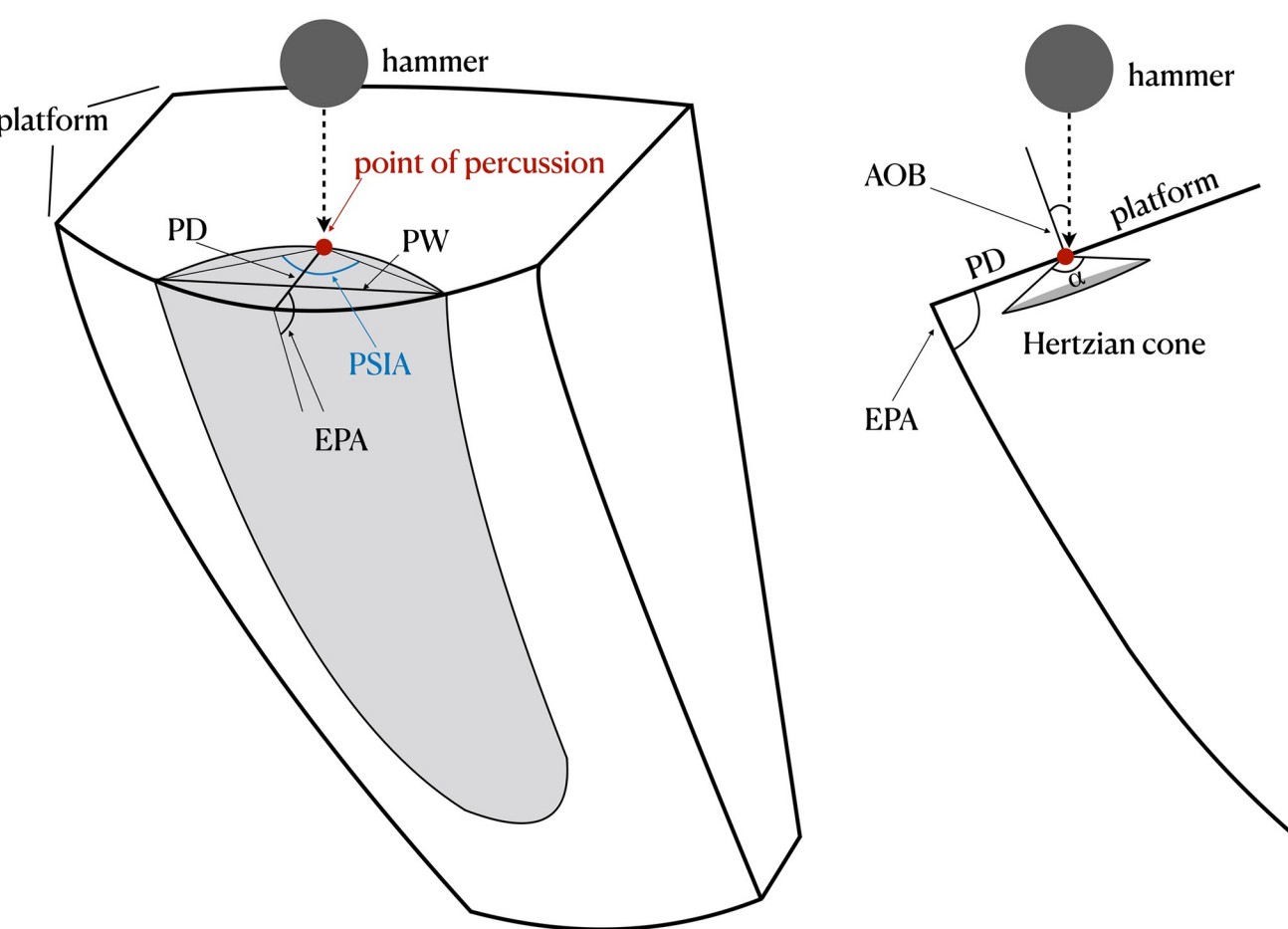

**Fig 1. Illustration of several flake attributes discussed here.** On the left is a schematic semispherical core used in the Dibble and colleagues glass experiments. On right is a schematic profile of the core through the point of percussion.

we start with the principle that the Hertzian cone, the angle of which is known from fracture mechanics to be a constant for a given raw material, has a measurable impact on flake formation.

When a core is struck, the force of impact begins to produce a Hertzian cone or "cone crack" at the point of percussion (see Fig 1B). The Hertzian cone is the characteristic feature of Hertzian fracture and is produced when a hard indenter is pressed onto the flat surface of a brittle solid [33–37]. In three dimensions it forms a truncated cone below the impact surface (see Fig 1B). In archaeology, Hertzian fracture is often referred to as conchoidal fracture. While the size of the Hertzian cone is dependent on variables such as the indenter's radius, the impact velocity and the fracture toughness of the brittle solid, the cone angle remains unchanged [38]. The Hertzian cone angle (sometimes referred to as the 'included angle' in fracture mechanics literature) is the apex angle of the Hertzian cone. Though the orientation of the Hertzian cone can be altered by changing the of angle of blow [36, 38–40], the apex angle remains constant for a given raw material type [41]. A number of fracture mechanics studies have demonstrated that the Hertzian cone angle is approximately 136° for soda lime glass [35–37], the type of glass used by Dibble and colleagues for their experiments. This value is also reported in Cotterell and Kamminga [2]. The cone crack grows as force is transferred to the core. Once the crack reaches a certain length, the crack's propagation path will no longer

be Hertzian. At this point, instead of propagating along the edges of the initial Hertzian cone, the crack continues almost parallel to the core surface to form a flake [1, 2]. Although the Hertzian cone is only associated with the initial crack formation, the angle of the cone leaves a marked effect on the ventral surface of the flake near where it has been struck in the form of the bulb of percussion.

Our prediction is that the constant Hertzian cone angle means that platform width of a given flake is a function of where the core is struck and where this angle intersects the platform edge. To show this we present a new measure called platform surface interior angle (PSIA) formed by the point of percussion and the extent of the platform width (Fig 1A). We predict that this angle will be constant given that it directly follows from the constant of the Hertzian cone angle. If our prediction is correct, platform width (PW) can be incorporated into the EPA-PD model and grounded in fracture mechanics via PSIA. This model also has behavioral implications in that it may explain how the manipulation of the platform impacts flake morphology.

To test this model, we examine several sets of flakes, including flakes produced in the Dibble glass experiments and flakes from replication experiments, using several methods to measure the PSIA. We find that the mean angle in all datasets, regardless of how it is measured, is the same (approximately 136 degrees) and quite consistent with above mentioned values for Hertzian cone formation [2]. There is some variability in the PSIA, and it is clear that this variability cannot be solely attributed to measurement error. In the Dibble glass experiments, where key variables are controlled, there is some indication that the PSIA responds to the angle of blow. Our finding is consistent with all of the empirical results of the Dibble experiments. More importantly, PSIA explains some of the patterns in those data that previously were unaccounted for. Once PSIA can be combined with the existing EPA-PD model, we may have a model for flake formation that can explain a larger portion of the variability we see in stone tool assemblages and that may allow for a closer link to predictions coming from fracture mechanics.

## Materials and methods

We examine the platform surface interior angle in three different datasets. First, we examine glass flakes (n = 142) and cores coming from the Dibble controlled experiments in flake formation [9, 19, 22, 24, 25]. This dataset has the advantage that a number of potentially important variables are either controlled for or were measured. These include the exterior platform angle, the angle of blow, the hammer type, raw material, and metrics such as platform thickness, platform width, flake length, width and thickness, and flake weight. Hereafter this dataset is referred to as the *Dibble glass data*. Second, we attempt to replicate the findings from the Dibble glass data by measuring the PSIA in a large (n = 568) set of complete, unretouched flakes coming from 45 discrete reduction sequences produced in replicative experiments by three knappers who were naïve to the goals of this study. These flakes were made with the intent of replicating various Middle and Upper Paleolithic core reduction strategies from the initial decortification of the core through to flake production and core maintenance [42–45]. These replicative experiments used nodules of high-quality Bergerac and Sénonien flint weighing from ~480–4100 g and coming from the southwest region of France [46]. For each of the flakes coming from these reduction series, the technology and the type of hammer (hard hammer, soft hammer and indirect percussion) are known. Here we are interested in the hard hammer flakes because these are comparable to the Dibble hard hammer flakes, but we have included the soft hammer and indirect percussion flakes in the results presentation for discussion purposes. Today these flakes are stored in Campagne, France, and hereafter this dataset is referred

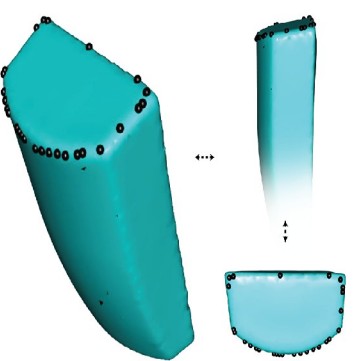
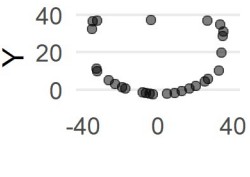
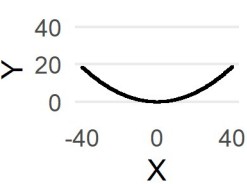

**Fig 2.** Left, an example of a semispherical glass core from the Dibble and colleagues experiments with landmarks on the platform outline. Above right, those same landmarks plotted in XY space. Below right, the core edge after fitting a polynominal curve to the XY landmarks.

to as the *Campagne data* (see [47] for additional details on the structure of this dataset). Third, in addition, we measured a small set of flint flakes (n = 67) produced at the Max Planck Institute for Evolutionary Anthropology in Leipzig, Germany, in the context of teaching, replication, and experimentation. In this case, no details are known about how the flakes were produced (e.g. hammer types). This set of flakes is used here only to test a method for measuring the PSIA. Hereafter this dataset is referred to as the *MPI data*.

The methods used to measure the PSIA varied substantially between the three datasets for practical reasons given the types of data available to us and because we wanted to try different methods to begin to have some idea of how best to measure this angle in future studies. First, for the Dibble glass data, we used the following procedure. Dibble and colleagues used several core forms, but the first and most common type is what was called the semispherical core. This core (reproduced here in Fig 2) looks like a loaf of bread with flat, squared off sides and back, and a curved or domed flaking surface. An unworked example of this core type was scanned using an NextEngine 3D laser scanner. The resulting mesh was then processed in R to rotate the platform to be perpendicular to the Z axis (or coincident with the XY plane). The XY coordinates of the triangles forming the platform were then extracted from this model and a convex hull fit to this cloud of points to have the full outline of the platform on the Dibble semispherical cores (see Fig 2). Next, we extracted just the portion of the outline that includes where flakes are struck from these cores, and we fit a polynomial curve to these points. Using the formula for this curve, we created a series of equally spaced (in X) points along the platform edge (see Fig 2). We then filtered the Dibble glass data to have only flakes made from the semispherical cores by hard hammer. We include only flakes with a feather termination, and we exclude flakes coming from experiments on platform beveling [25] and so-called 'on-edge' core strikes [19] (thereby arriving at our sample of n = 142). Knowing that Dibble and colleagues tried to strike flakes from these cores at the center or peak of the core surface curvature, we use the platform depths reported for these flakes to position the point of percussion relative to the set of platform edge outline points described above. Next, we find the symmetrical pair of platform edge outline points, one to the left and one to the right of the point of percussion, that yield a platform width equal to the reported platform width for each flake. Finally, the PSIA is calculated (arc-cosine of the dot product of two normalize vectors) as the angle between the two line segments formed by the left platform width point and the point of percussion and the right platform point and the point of percussion. In the results presented below, this angle is

referred to as the *estimated* PSIA to indicate that this angle is not directly measured from the flakes themselves.

We note that there are several potential sources of error in this method. First, we are assuming that each flake was struck from the center of the core. While this was the intention in the glass experiments, there is certainly some error associated with this. Second, we are assuming that the flake fracture plane is parallel to the core surface and not twisted towards one lateral side or the other. To the extent that either of these assumptions is invalid, it will impact the angle calculation.

To verify the angles computed in this way from the Dibble glass data, we also measure this angle directly with digital calipers and a goniometer on a subset of these flakes. We use two methods of measurement to begin to test how best to measure the PSIA by hand. In the first method, we measure the three sides of the triangle formed by the two platform width points and the point of percussion using digital calipers precise to.01 mm. Using standard trigonometric formulas, we then calculate the interior angle of this triangle that corresponds to the PSIA as described above. In the second method, we use a digital goniometer precise to 0.1 degrees to record this angle. The joint of the goniometer is positioned at the point of percussion and the jaws positioned to cross the two platform width points. Both of these methods come with possibilities for measurement error impacted by one's ability to pinpoint the point of percussion. In the Dibble glass flakes, because the core edge is standardized, identifying the two platform width points is fairly straightforward. However, in the goniometer method, taking the measurement to these points while avoiding the curvature of the bulb of percussion is not without some difficulties.

For the Campagne data, we use the following procedure. All of the flakes were scanned using an Artec surface scanner. Each of the flake meshes was then landmarked (see [47] for additional details on the scanning and landmarking). For our purposes, three of these landmarks are important: the two points (left and right) where the interior platform intersects the core surface (i.e., the two ends of the platform width) and the point of percussion. These three points are analogous with the three points described above for computing the PSIA. This angle, therefore, can be once again computed using the dot product of these two line segments (specifically the arc-cosine of the dot product of the normalize line segments). However, there is an important difference in that, with the Dibble glass data, all computations are with two dimensional line segments, while in the Campagne dataset the line segments are in three dimensions. In the latter case, the angle is computed in a two dimensional plane that is coincident with both line segments, but we note this difference because it could introduce a certain amount of incomparability in the two datasets. Our expectation is that these angles could average larger than the Dibble glass data because, for instance, lifting the point of percussion relative to the two platform points would result in a larger PSIA.

Lastly, for the MPI data, we use only the goniometer method described above. One of us (MW) made the measurements with instructions only on the mechanics of the measurement. To avoid bias, MW was naïve to the goals and results of this study. In the course of measuring the flakes, several problematic platforms were identified where the measurement of the PSIA was not as clear as the person selecting the flakes (SPM) had initially believed. These flakes were removed from the analysis.

We use the R [48] statistical environment to do this analysis. This paper is an rMarkdown document, and it is included in the supplementary information along with the data files needed to compile the document and replicate all of the figures, tables, and statistics. No permits were required for the described study, which complied with all relevant regulations.

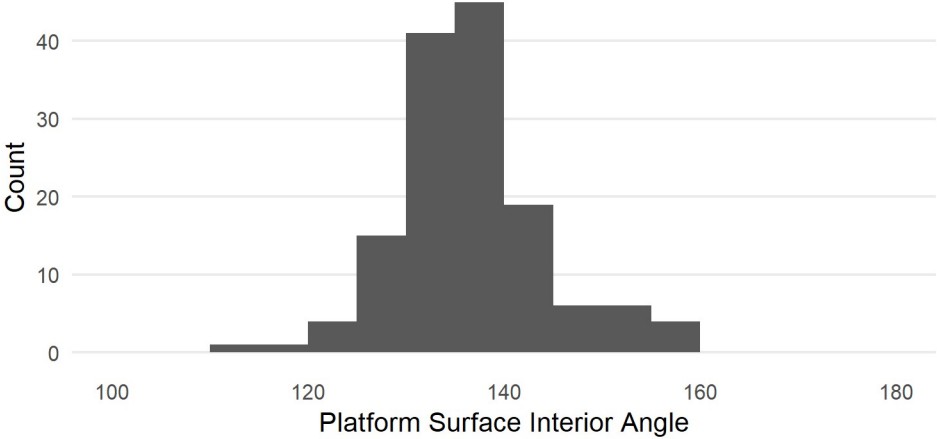

**Fig 3. Distribution of estimated PSIA based on the Dibble glass flakes.**

## Results

Fig 3 shows the distribution of estimated PSIA in the Dibble glass dataset. The distribution has a mean of 136.49 ± 7.56. Variation in this angle does not seem to be related to platform depth, exterior platform angle or weight (Fig 4). There is a relationship between platform width and the platform surface interior angle such that larger angles result in wider platforms, which is to be expected (see Fig 4).

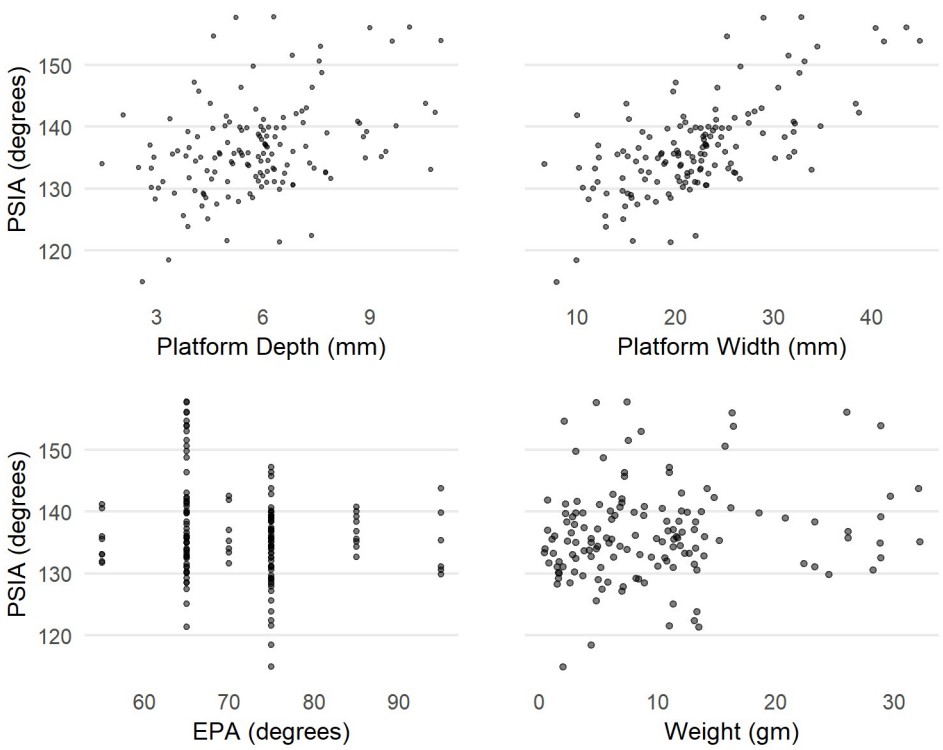

**Fig 4. PW, PD, EPA and weight against the estimated PSIA based on the Dibble glass flakes.**

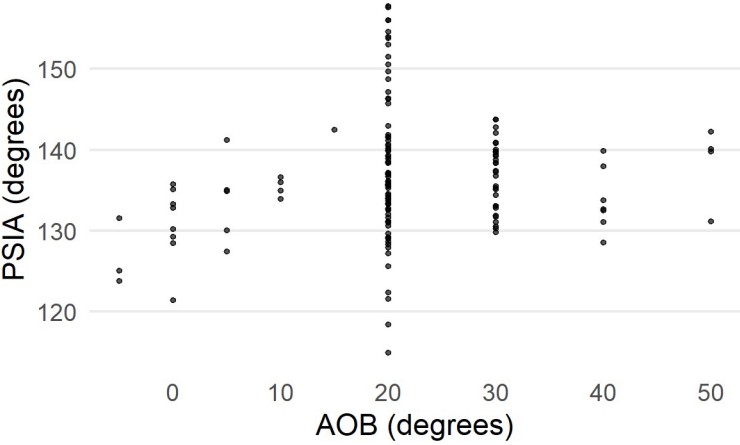

**Fig 5. AOB against the estimated platform surface interior angles based on the Dibble glass flakes.**

There also appears to be a relationship between the angle of blow and the PSIA (Fig 5). While there are fewer cases with angles of blow less than 20, there is some indication in the data that lower angles of blow may be correlated with lower PSIA. However, once an angle of blow is between 10-20 degrees or higher, the angle of blow is not correlated with the PSIA.

Another way of looking at the relationship between platform width and PSIA is to calculate what the platform width would be if the PSIA is a constant and compare this to the actual platform width. We can do this by placing the point of percussion on the same platform outlines as above using the known platform depth for each of the flakes in the Dibble glass data set. We then use the average PSIA computed above to extend two vectors from this point of percussion to the platform edge. Where these vectors intersect the platform edge defines the left and right limits of the platform width. This estimated value for platform width is then plotted against the actual, measured platform widths (Fig 6).

Figs 7 and 8 show comparisons of the results of the estimated PSIA presented above with direct measurements of this angle on a subsample of 49 of the 142 Dibble glass flakes in our analysis. For this sample, the PSIA is 135.71 ± 4.86. When measured with a digital goniometer

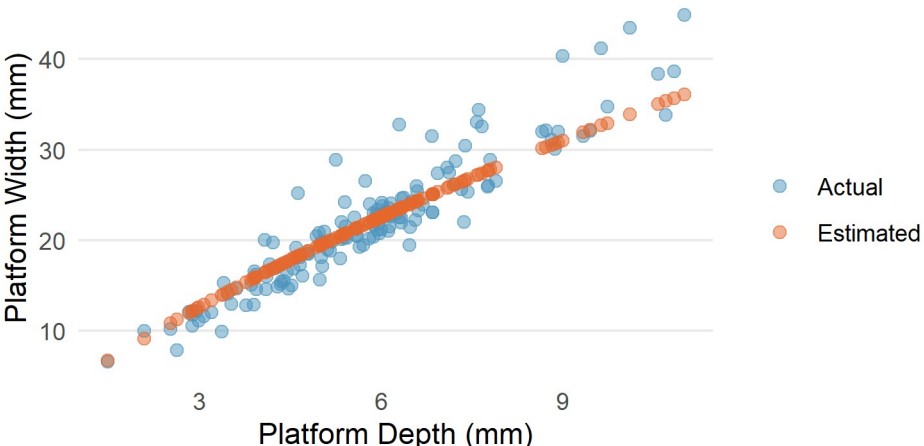

**Fig 6. The actual platform depth to platform width data from the Dibble glass core flakes and the estimated platform width using the average platform surface interior angle calculated previously.**

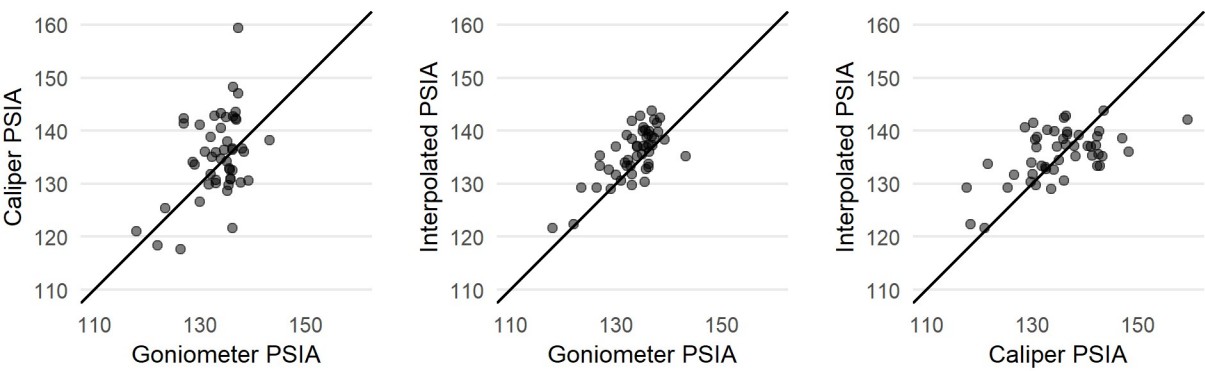

**Fig 7. A comparison of the estimated platform surface interior angle, this same angle as calculated from caliper measurements, and this same angle measured with a goniometer.** All measures are in degrees.

the angle is 133.44 ± 4.61, and when measured with digital calipers and calculated using trigonometry the angle is 135.86 ± 8.85.

The distribution of the PSIA for the 568 flakes in the Campagne dataset is shown in Fig 9. In the hard hammer flakes only, which is the technique used in our subset of the Dibble glass dataset, the mean platform surface interior angle is 140.36 ± 12.38. Because data are available for punch and soft hammer flakes, these are presented as well. Punch flakes have a lower PSIA (126.53 ± 11.01), and soft hammer flakes have a mean of 138.94 ± 15.69. The mean PSIA for all flakes in the Campagne data set is 138.63 ± 13.27.

In the Campagne data, PSIA does not covary with platform width, platform depth or the shape of the platform (as measured by the ratio of platform width to platform depth) (Fig 10). Though sample size is potentially a problem, there is perhaps an indication that for larger platform depths, there is less variability in the PSIA.

The distribution of PSIA in the MPI dataset as measured by a digital goniometer is presented in Fig 11. This dataset contains 67 flakes, and the mean is 137.75 ± 10.97, in keeping with the other datasets.

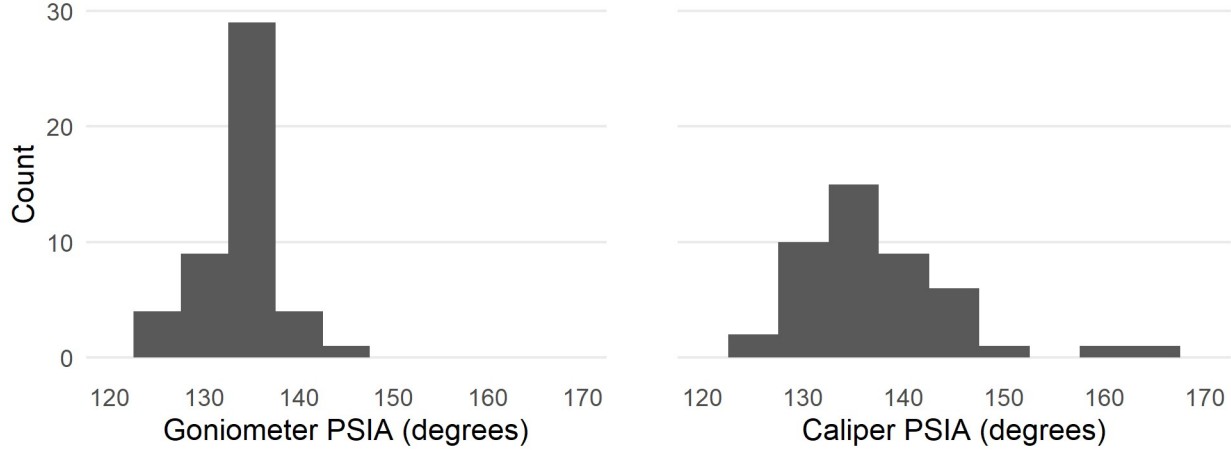

**Fig 8. The distributions of platform surface interior angle as directly measured with a goniometer and as calculated from caliper measurements.**

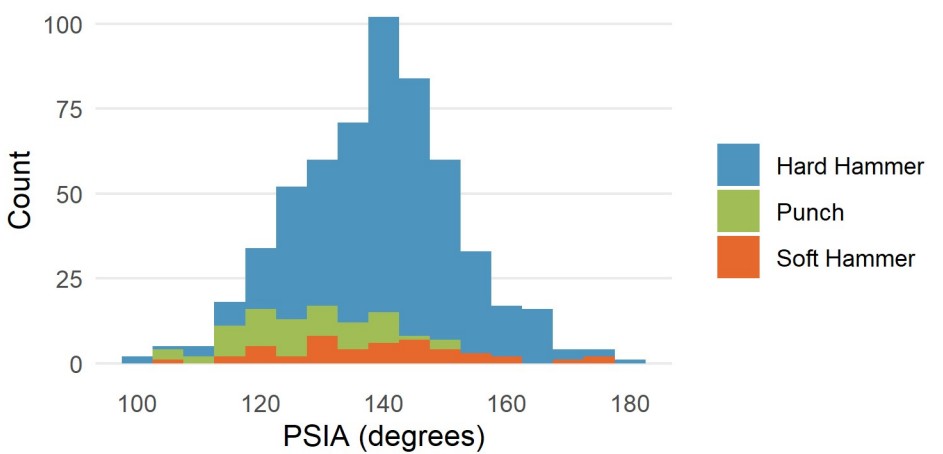

**Fig 9. Distribution of PSIA in the Campagne data set color coded by percussion type.**

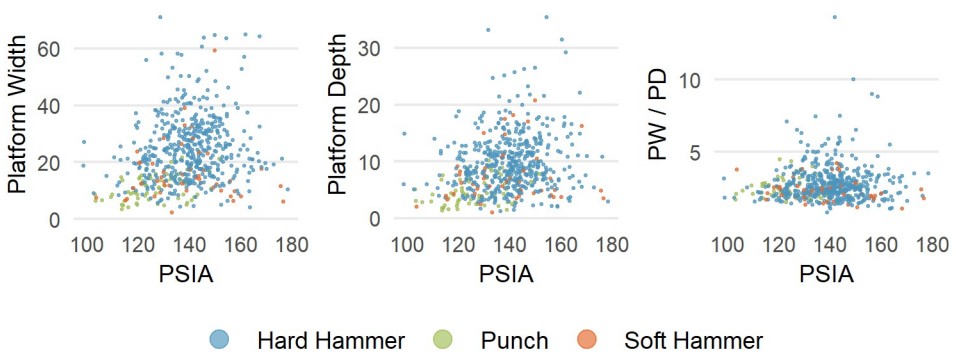

**Fig 10. PSIA as a function of platform width, platform depth, and EPA in the experimental flake collection.** Color coding is the same as in Fig 8.

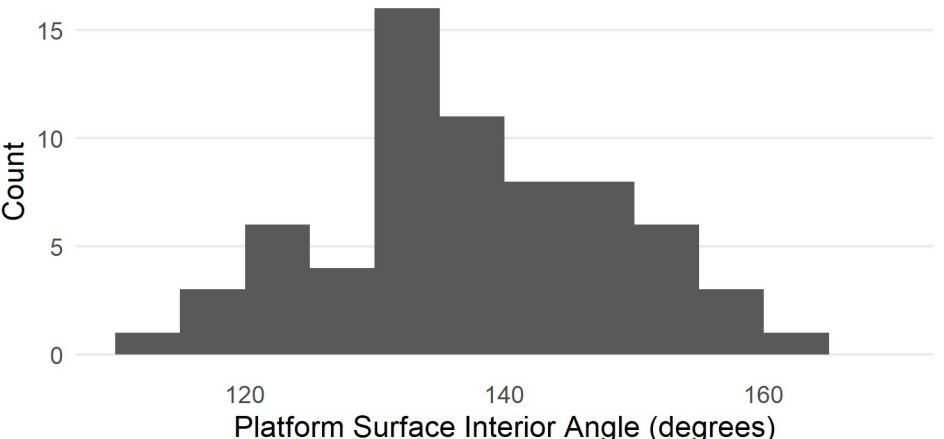

**Fig 11. Distribution of platform surface interior angle in the MPI flakes dataset as measured by a digital goniometer.**

## Discussion

Each of the datasets and measurement methods used to determine the PSIA yielded very similar results, and these results are consistent with the prediction based on the already known and constant angle of Hertzian cones in soda lime glass (136˚). While these results directly incorporate platform width into the EPA-PD model of flake formation, they also suggest that the existing model requires additional nuance. All other things being equal, the relationship between PSIA and platform width indicates that what determines the size of the flake is how far into the core it is struck. Specifically, because of the constant PSIA, the deeper into the core the flake is struck, the greater the platform width will become (and the thicker the flake will become as well). However, and this is the nuance, while platform depth is normally a good proxy for how far into the core the flake is struck, it is not exactly the same thing. What platform depth really measures is how far into the *existing platform* of the core the flake is struck. Beveled flakes illustrate this point.

Beveled flakes are ones where material is removed behind the platform prior to striking the core. Dibble and colleagues recognized that beveling altered the EPA-PD model of flake formation such that the interaction of platform depth and exterior platform angle no longer predicted flake size [25]. Beveled flakes have too thin a platform for their size. However, PSIA helps explain this discrepancy and thereby pulls beveled flakes back into an expanded EPA-PD model. To illustrate this point, we examine some beveled flakes in the Dibble glass dataset.

In the Dibble glass dataset there are 11 flakes with flat or concave beveling, coming from semispherical cores, and otherwise conforming to our selection criteria. These are plotted in Fig 12 along with the non-beveled flakes. It is clear that beveling changes the relationship between platform depth and platform width. For a given platform width, the beveled flakes have much shallower platforms (smaller PD) than expected. As a result, the EPA-PD model underestimates their weight (see Fig 12) as well. To illustrate the power of PSIA to model these flakes, we build a linear model to estimate platform depth from the PSIA and platform width in the non-beveled flakes in the Dibble glass dataset. We then use this model to predict a platform depth for the beveled flakes. However, given that the linear model requires PSIA to predict platform depth and the actual PSIA is not known for these flakes, we substitute in its place the average PSIA, as reported above for the non-beveled flakes. When this is done, the predicted platform depths for the beveled flakes plot on the same trend line as the non-beveled flakes (compare red and blue points in Fig 12).

This modeled platform depth is then be used to improve the EPA-PD model to give better estimates of flake size. The main aspect of size that Dibble and colleagues have focused on with the EPA-PD model is weight, and so we model flake weight as a function of EPA, platform depth and the interaction of the two (see Fig 12). The cube root of weight is used to correct for the different dimensions in the model. Next, we use this same model to predict flake weight in the beveled flakes using the platform depth as originally measured on these beveled flakes. In this case, the modeled flake weights are much too low in comparison to their actual weights (red points in Fig 12). Finally, we use the predicted platform depth for the beveled flakes, as modeled above, to predict flake weight again using the non-beveled flake model. In this case, the flake weights plot in among the rest of the non-beveled flakes (blue points in Fig 12). Thus the beveled flakes are the expected size when we think of PD in the EPA-PD model not as a measure of platform depth but rather as a measure of how far into the core the flake is struck, which then determines the flake width via the PSIA given the shape of the platform edge. Beveling does not change the expected size of these flakes when flake formation is viewed this way.

There is some indication in the Dibble glass data that the angle of blow may impact the PSIA. As mentioend earlier, at low angles of blow (with 0 degrees representing an angle of

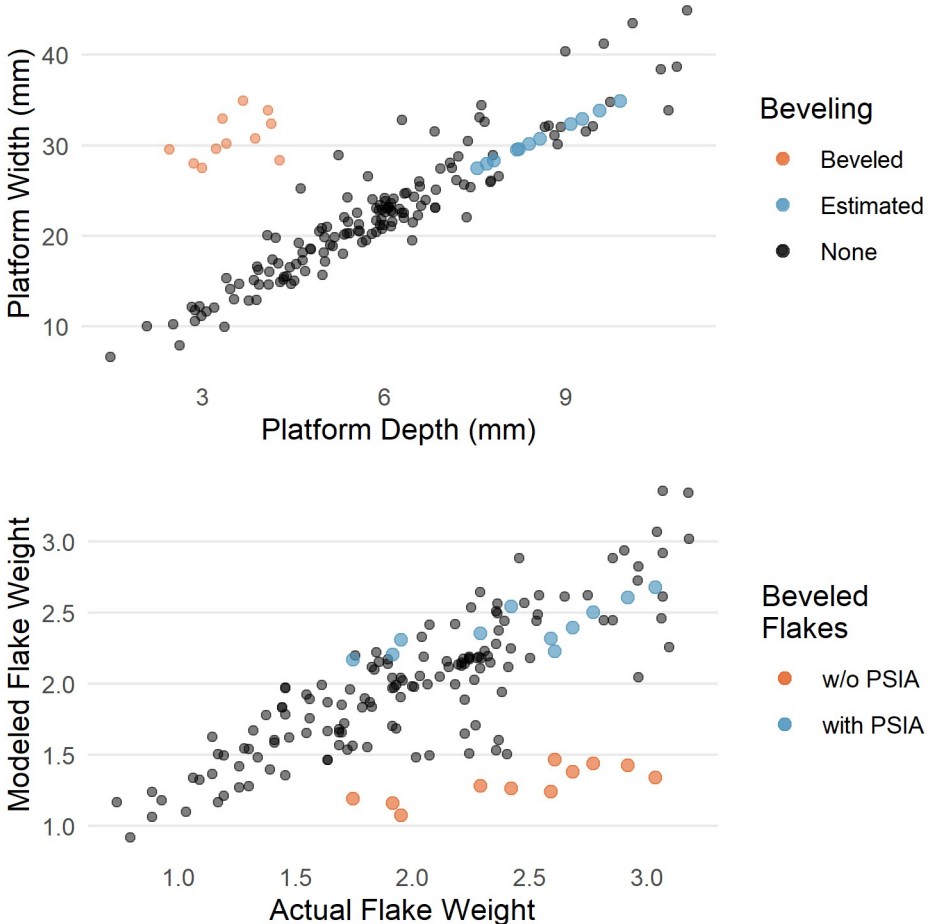

**Fig 12. Platform depth to platform width including beveled flakes (red) and non-beveled flakes (black) (top).**
Estimated points in blue are the same beveled flakes but with the platform depth predicted using the average PSIA and their actual platform width. At the bottom, the actual flake weight is compared to the predicted flake weight based on an EPA-PD model for non-beveled flakes (points in black). The predicted weight using the actual (w/o PSIA) and the modeled (with PSIA) platform depths for the beveled flakes are then plotted as well in red and blue respectively.

blow perpendicular to the platform, see Fig 2 in [19]), the PSIA is below average, and it appears to increase until the angle of blow reaches between 10 and 20 degrees (from perpendicular) after which the PSIA remains essentially unchanged (see Fig 5). With the caveat that the Dibble glass data set has very few cases with angles of less than 20 degrees, the fracture mechanics literature suggests a relationship like this: the angle of blow changes the direction, though not the size, of the Hertzian cone such that flakes struck from cores with a high angle of blow (oblique strike) should have "steeper and less prominent cones and less salient bulbs of percussion than flakes which are struck more steeply [or more perpendicular]" ([4], page 38). Experimentally, Magnani et al. [19] also find a relationship between these variables. In their case, a negative angle of blow (here values less than 0 meaning a strike directed into the interior of the core rather than towards the core surface) results in smaller bulbs relative to the weight of the flake. In our Campagne data set, we see a difference between flakes made from direct hard-hammer percussion and those made with a punch technique. The latter cluster at the low range of PSIA. This difference could be interpreted as reflecting a difference in the angle of blow in that punch flakes are more likely to be struck perpendicularly to the platform surface (an angle of blow of 0). More work needs to be done, in particular in analyzing the Dibble glass data set

where angle of blow is well controlled, but we suggest that increasing the angle of blow has the effect of tipping the direction of the Hertzian cone such that it intersects the core surface not as a circle but rather as an ellipse. This is a phenomenon that has been repeatedly documented in fracture mechanics [36, 38, 40]. Although the angle of the cone itself remains unchanged, its intersection with the surface broadens and results in higher PSIA. Thus, if this suggestion is correct, striking a core with a high angle of blow will result in a larger platform width for a given platform thickness.

The direct measurement of PSIA in a subsample of the Dibble glass flakes shows that our method for finding this angle using the platform surface shape and platform depth is working. However, there is variability in this angle depending on how it is measured. With the caveats that only one person measured these flakes and that the sample is small, in general it seems that the direct measurement with a goniometer performs better than the indirect calculation of the angle from three caliper measurements. Of the two methods, the caliper measurements show greater variability than do the goniometer measurements. The caliper method can also fail completely when measurement error produces a triangle with impossible side lengths (e.g. the sum of the two shorter sides is less than the length of the longest side). Importantly too, this is only knowable once the measures are taken and an angle is calculated, making it much more difficult to correct, whereas the goniometer method produces an angle each time. Our study, however, does not indicate which of the three methods in this case is correct, nor do we know the error associated with any of these individual methods. Now that PSIA seems to produce results relevant to understanding flake formation, additional studies are required to better know the error distribution on an individual measure. We note too that this error distribution will likely vary with the angle itself, the size of the point of percussion, and other factors that remain to be identified. The question, however, is whether this measurement error will overwhelm patterns, such as, for instance, showing a potential correlation in angle of blow and PSIA. It is our expectation that with a digital goniometer, direct measurement of the PSIA can become one of the standard measurements within lithic attribute analysis, but this remains to be determined.

We note that our finding that the PSIA varies around a constant derived from fracture mechanics appears to be consistent with all of the findings to date of the Dibble glass experiments [9, 19, 22, 24, 25]. Additionally, it perhaps helps explain or account for one of the more counter-intuitive findings of the glass experiments, namely that flake size (weight) is not impacted by the force with which the core is struck [9]. In the EPA-PD model, the amount of force required to remove a flake given a particular combination of EPA-PD is a constant (see also [20]). Subtracting force means that the flake is not initiated. Adding force does not change the size of the resulting flake. This makes sense in the PSIA addition to the model. In fracture mechanics, it is known that striking a material harder will change the size of the Hertzian cone but not the angle [35]. Thus when a given core is struck, how far into the core the Hertzian cone can travel will depend on the hammer size and the striking force, but where the cone will intersect the core surface does not change. So striking a Dibble glass core harder at a particular point does not change the platform width, and as a result, the subsequent fracture plane that removes the flake has much less freedom to change the size of the flake. This said, we note that a recent study [40] shows that increasing impact velocity of the indenter will eventually cause the Hertzian cone angle to decrease (see also [49]). The question is whether these velocities are relevant to stone knapping.

## Conclusions

Fracture mechanics is a massive field of study with both great potential and great difficulties for understanding flake formation. The potential is that the physical laws and models coming

from fracture mechanics are, ultimately, how the actions of knappers are translated to the removal of usable flakes. The difficulties are both inherent to the field of fracture mechanics itself and to the complexity of the problem (some solutions require more time and computing power than exists). There are also difficulties related to the challenges of interdisciplinary work where the equations and goals of one field are terribly difficult to bring into another. This later point is clearly seen in the early fracture mechanics literature and the minimal impact it has had on experimental and replicative studies of flake formation.

This said, our goal here was to return to this literature and to try to find some useful insights that could be translated to a better understanding of the underlying mechanisms (or first principles) of flake formation and that might thereby lead to a better integration of the various statistical models currently in use. To do this, we switched from the existing focus in the experimental literature on what knappers do to instead focus on attributes that may be more directly related to fracture mechanics. Thus we focused our attention on the Hertzian percussion cone as a constant and investigated whether it could help explain platform width, an aspect of flake size and shape that up to now has been absent from the dominant EPA-PD model of flake formation. We measured three different collections in multiple different ways and found that in each case the angle formed by the platform width and the point of percussion to be, on average, essentially the same as what is predicted from fracture mechanics for the angle of the Hertzian cone.

We conclude that the platform surface interior angle or PSIA is an important determinant in flake size and shape. While it would seem that it is a constant and not under direct control by the knapper, the knapper is (unknowingly) exploiting the properties of this angle when preparing the platform and its contact with the core surface and when deciding how far into the core to strike. In fact, this angle underlies a knapper's ability to visualize the extent of the striking platform prior to removing a flake. In our model, this is how the PD side of the EPA-PD model of flake formation is translated into a flake of a particular size and shape. In other words, it is not the PD that directly structures flake size but rather PD is proxy for how far into the core a flake is struck which then, through the effect of platform surface interior angle on platform width, structures the size and shape of the flake.

This, however, requires further testing, particularly with beveled flakes where PD, without PSIA, performs poorly as a proxy for how far into the core a flake is struck. If the PSIA performs better in these circumstances, as the preliminary data presented here suggest (see Fig 11), then we will have improved the EPA-PD model and helped to integrate what are now disparate studies [9, 25]. Additionally, while our study shows that the average PSIA conforms well to predictions from fracture mechanics for Hertzian flake formation, it is still clear that there is variability around this mean. Given that there is certainly some chaos in flake formation, we are not sure how much variability to expect. For instance, sometimes the Hertzian cone crack may kink at a light angle to continue expanding, and this kink may contribute to the variability observed in PSIA [50, 51]. However, it is also clear that the model does not work at all for some flakes. It may be the case that these flakes are not formed by Hertzian mechanisms (e.g. bending flakes) and that alternative models will be required in these cases. Clearly more experiments and more data are required to begin to understand which kinds of flakes fail the PSIA model presented here, and these flakes will require additional insights into models of flake formation.

## Supporting information

**S1 File. rMarkdown file of this document.**
(RMD)

**S2 File. Compressed file containing data files, figures, bibliography and format file.**
(ZIP)

## Acknowledgments

Michel Brenet and Laurence Bourguignon produced the Campagne data set, and we thank them for allowing us access to this important collection. Claudio Tennie provided feedback on the ideas and comments on the manuscript. SPM, MW, WA, ZR and TD thank Jean-Jacques Hublin for his continued support of our research agenda. Sadly Harold Dibble died before the main findings of this paper were discovered. Without his vision and his efforts to build a data set of flakes made under controlled conditions, this paper would not have been possible. We dedicate this paper to him.

## Author Contributions

**Data curation:** Aylar Abdolahzadeh, Will Archer, Annie Chan, Igor Djakovic, Tamara Dogandžić, George M. Leader, Li Li, Sam Lin, Matthew Magnani, Zeljko Rezek, Marcel Weiss.

**Methodology:** Shannon P. McPherron.

**Writing – original draft:** Shannon P. McPherron, Li Li, Sam Lin, Jonathan Reeves, Zeljko Rezek.

**Writing – review & editing:** Shannon P. McPherron, Aylar Abdolahzadeh, Will Archer, Annie Chan, Igor Djakovic, Tamara Dogandžić, Li Li, Sam Lin, Jonathan Reeves, Zeljko Rezek, Marcel Weiss.

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
