## [Decision Letter · Decision Letter 0]

30 Jul 2020

PONE-D-20-16801

Introducing Platform Surface Interior Angle (PSIA) and Its Role in Flake Formation, Size and Shape

PLOS ONE

Dear Dr. McPherron,

Thank you for submitting your manuscript to PLOS ONE. After careful consideration, we feel that it has merit but does not fully meet PLOS ONE’s publication criteria as it currently stands. Therefore, we invite you to submit a revised version of the manuscript that addresses the points raised during the review process.

I received comments from two reviewers. Their comments are attached.Please address these comments carefully and significantly improve both English and presentations of this manuscript.

We look forward to receiving your revised manuscript.

Kind regards,

Jianguo Wang, PhD

Academic Editor

PLOS ONE

Journal Requirements:

2. In your manuscript, please provide additional information regarding the specimens used in your study. Ensure that you have reported specimen numbers and complete repository information, including museum name and geographic location.

For more information on PLOS ONE's requirements for paleontology and archaeology research, see https://journals.plos.org/plosone/s/submission-guidelines#loc-paleontology-and-archaeology-research.

"We thank the Max Planck Society for funding portions of this work. LL and JR received funding from the

European Research Council (ERC) under the European Union's Horizon 2020 research

and innovation program (grant agreement n° 714658; STONECULT project). TD

received funding from the European Union's Framework Programme for Research and

Innovation Horizon 2020 (2014-2020) under the Marie Sk lodowska-Curie grant

agreement No. 751125SPM, MW, WA, ZR and TD thank Jean-Jacques Hublin for his

continued support of our research agenda."

Reviewers' comments:

Reviewer's Responses to Questions

**Comments to the Author**

1. Is the manuscript technically sound, and do the data support the conclusions?

Reviewer #1: Yes

Reviewer #2: Yes

2. Has the statistical analysis been performed appropriately and rigorously? 

Reviewer #1: Yes

Reviewer #2: Yes

3. Have the authors made all data underlying the findings in their manuscript fully available?

Reviewer #1: Yes

Reviewer #2: Yes

4. Is the manuscript presented in an intelligible fashion and written in standard English?

Reviewer #1: Yes

Reviewer #2: No

5. Review Comments to the Author

Reviewer #1: The paper by McPherron et al introduces PSIA- Platform Surface Interior Angle as a new quantifiable variable to understand fracture mechanics in flake formation, shape, and size variability through statistical modelling. This study adds a constant to the EPA-DP model and the study of fracture mechanics to explain flake variability.

Though there is a need for further testing and data, which is clearly stated by the authors, the paper sets the principles and methodology to achieve it and expand it towards a comprehension of flake formation.

I believe it will be a reference for a growing research field and particularly important towards the development of a general model for understanding technological variability.

Only minor revisions are recommended:

Page 5 (103-105): the word “majority” can be subjective. Perhaps following the argument with supporting references would enrich the paragraph.

Page 8 (179-184): Though the paper does not centre on raw material variability, the description criteria for the three assemblages used in the study should be homogenous. In materials and methods, the description of Dibble glass data and Campagne data refer to the raw material used. For the MPI data, this information is missing.

Page 8 (200): I assume the filtered Dibble glass data is represented by 49 samples referred on page 11 - 288. To make it easier for the reader to follow the results, the number of samples should be included earlier in the description of materials and methods.

Figure 10: it is recommended to add labels in figure 10. Though the same labels are present in figure 9, it makes it easier for the reader to have the corresponding information in each graph or figure.

Reviewer #2: In the present manuscript, the linkage between fracture mechanics and the results obtained from controlled experiments is explored. By documenting the influence of Hertzian cone formation, the authors found that the platform width is a function of the Hertzian cone constant angle and the geometry of the platform edge. In the referee’s opinion, the formulation and method of this paper should be right, but the following revision should be carefully made: Please find the review comments in the attached file

6. PLOS authors have the option to publish the peer review history of their article (what does this mean?). If published, this will include your full peer review and any attached files.

Reviewer #1: No

Reviewer #2: No

---

## [Author Response · Author response to Decision Letter 0]

10 Sep 2020

[see uploaded response document]

Response Document

**Reviewer #1**: The paper by McPherron et al introduces PSIA- Platform Surface Interior Angle as a new quantifiable variable to understand fracture mechanics in flake formation, shape, and size variability through statistical modelling. This study adds a constant to the EPA-DP model and the study of fracture mechanics to explain flake variability.

Though there is a need for further testing and data, which is clearly stated by the authors, the paper sets the principles and methodology to achieve it and expand it towards a comprehension of flake formation.

I believe it will be a reference for a growing research field and particularly important towards the development of a general model for understanding technological variability.

Only minor revisions are recommended:

Page 5 (103-105): the word “majority” can be subjective. Perhaps following the argument with supporting references would enrich the paragraph.

**Whew. Not sure how to address this one. We agree that it is a subjective statement without citation, but we also think that it would be difficult to develop this point without at least an additional paragraph where we would have to call out individual studies in a somewhat negative way. This, we think, would distract from our paper. What we have done is reworded a bit and added a citation to Speth 1972 where he makes the same point decades ago.**

Page 8 (179-184): Though the paper does not centre on raw material variability, the description criteria for the three assemblages used in the study should be homogenous. In materials and methods, the description of Dibble glass data and Campagne data refer to the raw material used. For the MPI data, this information is missing.

**We added that the flakes are made from flint.**

Page 8 (200): I assume the filtered Dibble glass data is represented by 49 samples referred on page 11 - 288. To make it easier for the reader to follow the results, the number of samples should be included earlier in the description of materials and methods.

**There is a confusion here and we modified the text to make this more clear. First, we restate the Dibble glass sample size (142) when discussing the filtering. Second, no, the 49 figure is a subsample of the 142. We now call this a subsample and we restate the sample sizes**.

Figure 10: it is recommended to add labels in figure 10. Though the same labels are present in figure 9, it makes it easier for the reader to have the corresponding information in each graph or figure.

**Done.**

**Reviewer #2**

Comments on PONE-D-20-16801 

In the present manuscript, the linkage between fracture mechanics and the results 

obtained from controlled experiments is explored. By documenting the influence of 

Hertzian cone formation, the authors found that the platform width is a function of the 

Hertzian cone constant angle and the geometry of the platform edge. In the referee’s 

opinion, the formulation and method of this paper should be right, but the following 

revision should be carefully made: 

1. The drawings in the paper are bad, please improve them. For example, the present 

setup presented in Fig.1 is unclear. 

**We redrew the figure to make it more clear and slightly better in appearance. We note, however, that this figure is in keeping with similar figures in the literature.** 

2. There is something wrong with the citation in the paper, for example, no issue 

number for Reference 17. 

**This particular publication does not have an issue number. However, pages were missing, and we corrected that. We also went through all other citations and checked them for accuracy and missing information. Several were corrected. We also updated our citation to R.**

3. There are plenty of improper expressions in the manuscript, for example, 

“Increasing either increases flake size, but the relationship between the two is 

geometric such that at higher values of EPA changes in PD have a greater effect on 

flake size.” 

**We modified this sentence to increase clarity.**

“The EPA-PD model of flake formation, however, is constrained in what it can 

explain.” 

**We think this sentence is clear, especially in the context of what follows.**

“In particular, we start with the principle that the Hertzian cone, the angle of which is 

known from fracture mechanics to be a constant for a given raw material, has a 

measurable impact on flake formation.” 

**We struggle to find a way to say this more clearly.**

“While the size of the Hertzian cone is dependent on variables such as the indenter's 

radius, impact velocity and fracture toughness of the brittle solid, the cone angle 

remains unchanged” 

**We modified this sentence slightly.**

“This dataset has the advantage that a number of potentially important variables are 

either controlled for or were measured.” 

**We struggle to find a way to say this more clearly.**

“In the results presented below, this angle is referred to as the estimated PSIA to 

indicate that is is not directly measured from the flakes themselves.” 

**Fixed.**

Please check the manuscript carefully and improve them. 

**We have done this.**

4. More explanation about the EPA-PD model and PSIA will be more useful for 

readers of the present paper. 

**We are not sure what else we can say about the EPA-PD model, and we have provided extensive citations to the original papers defining this model plus papers applying it. And given that we are introducing PSIA for the first time and that the paper is entirely about PSIA, we are also not sure what more explanation we can give. We want to emphasize to the reviewer that we do not understand the mechanics behind these models as well as we would like. This paper tries to tie PSIA to Hertzian cone formation, but exactly how that happens is unclear (not just to us). These will be things that additional research can continue to clarify. **

5. The reviewer suggests the authors to add detailed information about the solution 

procedure in the present paper. 

**We are not sure what a solution procedure is, and so we Googled it. If we understand correctly from the examples we found, this is well beyond our expertise to do. We also doubt we have a sufficient understanding of the mechanisms involved at this point to do such a thing. While not exactly the same thing, we do note that all of our code and all of our data are included with this paper.**

6. Whether the present model is related to contact behavior of an indenter on a 

substrate? If so, some related references should be helpful, for example, J. Appl. 

Mech.-Trans. ASME. 2015. 82(4): 041008. Int. J. Solids Struct. 2013. 50(7): 1108-1119. Int. J. Mech. Sci. 2019. 

151: 410-423. 

**We thank the reviewer for drawing our attention to these papers. As we further investigate different strike conditions, we think this literature could be valuable. It certainly points to issues we will need to consider. For the present paper, however, we think that this discussion on our part would be too speculative at this time and draw away from our main point.**

---

## [Decision Letter · Decision Letter 1]

20 Oct 2020

Introducing Platform Surface Interior Angle (PSIA) and Its Role in Flake Formation, Size and Shape

PONE-D-20-16801R1

Dear Dr. McPherron,

We’re pleased to inform you that your manuscript has been judged scientifically suitable for publication and will be formally accepted for publication once it meets all outstanding technical requirements.

Kind regards,

Jianguo Wang, PhD

Academic Editor

PLOS ONE

Additional Editor Comments (optional):

Reviewers' comments:

Reviewer's Responses to Questions

**Comments to the Author**

1. If the authors have adequately addressed your comments raised in a previous round of review and you feel that this manuscript is now acceptable for publication, you may indicate that here to bypass the “Comments to the Author” section, enter your conflict of interest statement in the “Confidential to Editor” section, and submit your "Accept" recommendation.

Reviewer #1: All comments have been addressed

Reviewer #2: All comments have been addressed

2. Is the manuscript technically sound, and do the data support the conclusions?

Reviewer #1: Yes

Reviewer #2: Yes

3. Has the statistical analysis been performed appropriately and rigorously? 

Reviewer #1: Yes

Reviewer #2: Yes

4. Have the authors made all data underlying the findings in their manuscript fully available?

Reviewer #1: Yes

Reviewer #2: Yes

5. Is the manuscript presented in an intelligible fashion and written in standard English?

Reviewer #1: Yes

Reviewer #2: Yes

6. Review Comments to the Author

Reviewer #1: The authors made the proposed corrections and responded adequately to all comments and queries raised in the first round of revisions. Considering changes made throughout the text I have no new comments to ad regarding the content of the article. I propose the manuscript to be accepted for publication.

Reviewer #2: (No Response)

7. PLOS authors have the option to publish the peer review history of their article (what does this mean?). If published, this will include your full peer review and any attached files.

Reviewer #1: No

Reviewer #2: No

---

## [Editor Report · Acceptance letter]

9 Nov 2020

PONE-D-20-16801R1 

Introducing Platform Surface Interior Angle (PSIA) and Its Role in Flake Formation, Size and Shape 

Dear Dr. McPherron:

I'm pleased to inform you that your manuscript has been deemed suitable for publication in PLOS ONE. Congratulations! Your manuscript is now with our production department. 

Kind regards, 

on behalf of

Dr. Jianguo Wang 

Academic Editor

PLOS ONE